# Predictors of Renal Function Worsening in Patients with Chronic Obstructive Pulmonary Disease (COPD): A Multicenter Observational Study

**DOI:** 10.3390/nu13082811

**Published:** 2021-08-16

**Authors:** Corrado Pelaia, Daniele Pastori, Giuseppe Armentaro, Sofia Miceli, Velia Cassano, Keti Barbara, Giulia Pelaia, Maria Perticone, Raffaele Maio, Pasquale Pignatelli, Francesco Violi, Francesco Perticone, Giorgio Sesti, Angela Sciacqua

**Affiliations:** 1Department of Medical and Surgical Sciences, University Magna Græcia of Catanzaro, 88100 Catanzaro, Italy; pelaia.corrado@gmail.com (C.P.); peppearmentaro@libero.it (G.A.); sofy.miceli@libero.it (S.M.); velia.cassano@libero.it (V.C.); keti.b.93@gmail.com (K.B.); giulia.pelaia@gmail.com (G.P.); mariaperticone@hotmail.com (M.P.); raf_maio@yahoo.it (R.M.); perticone@unicz.it (F.P.); 2Department of Clinical Internal, Anesthesiological and Cardiovascular Sciences, Sapienza University of Rome, 00185 Rome, Italy; daniele.pastori@uniroma1.it (D.P.); pasquale.pignatelli@uniroma1.it (P.P.); francesco.violi@uniroma1.it (F.V.); 3Department of Clinical and Molecular Medicine, Sapienza University of Rome, 00161 Rome, Italy; giorgio.sesti@uniroma1.it

**Keywords:** COPD, uric acid, diabetes, renal failure, oxidative stress

## Abstract

Background. Chronic obstructive pulmonary disease (COPD) patients have multiple comorbidities which may affect renal function. Chronic kidney disease (CKD) is a risk factor for adverse outcomes in COPD patients. The predictors of CKD in COPD are not well investigated. Methods. A multicenter observational cohort study including patients affected by COPD (GOLD stages 1 and 2) was carried out. Principal endpoints were the incidence of CKD, as defined by an estimated glomerular filtration rate (eGFR) <60 mL/min/1.73 m^2^, and the rapid decline of eGFR >5 mL/min/1.73 m^2^/year. Results. We enrolled 707 outpatients. Overall, 157 (22.2%) patients had CKD at baseline. Patients with CKD were older, with higher serum uric acid (UA) levels, and lower FEV_1_. During a mean follow-up of 52.3 ± 30.2 months, 100 patients developed CKD, and 200 patients showed a rapid reduction of eGFR. Multivariable Cox regression analysis displayed that UA (hazard ratio (HR) 1.148, *p* < 0.0001) and diabetes (HR 1.050, *p* < 0.0001) were predictors of incident CKD. The independent predictors of rapidly declining renal function were represented by an increase of 1 mg/dL in UA (odds ratio (OR) 2.158, *p* < 0.0001)), an increase of 10 mL/min/1.73 m^2^ in baseline eGFR (OR 1.054, *p* < 0.0001) and the presence of diabetes (OR 1.100, *p* < 0.009). Conclusions. This study shows that COPD patients have a significant worsening of renal function over time and that UA and diabetes were the two strongest predictors. Optimal management of these risk factors may reduce the incidence of CKD in this population thus probably improving clinical outcome.

## 1. Introduction

Chronic obstructive pulmonary disease (COPD) is a widespread systemic inflammatory disorder, characterized by persistent respiratory symptoms and airflow limitation, which often is not fully reversible [1]. COPD results from a complex interplay between genetic susceptibility and exposure to environmental stimuli, including tobacco smoke and air pollution [2]. COPD represents a major cause of mortality and morbidity, and its incidence is going to increase in the next years. The individual burden associated with COPD, as well as its health costs, are strongly correlated with a wide spectrum of cardiovascular (CV) and non-cardiovascular comorbidities such as hypertension, diabetes, cognitive decline, osteoporosis/osteopenia, chronic kidney disease (CKD) and cancer [3,4].

In particular, COPD patients have a higher risk of developing CKD than the general population [5], with age, diabetes, arterial hypertension and overweight being the most common risk factors for new onset of CKD [6]. The pathogenesis of CKD includes atherosclerotic damage, due to activation of proinflammatory and pro-oxidant pathways leading to pathologic changes in renal circulation [7,8,9]. Within this context, interleukin-6 (IL-6) is the best-known cytokine that, produced by pulmonary cells, reaches the systemic circulation and targets other organs, thus exerting further inflammatory effects. In addition to IL-6, also interleukin-1β (IL-1β), interleukin-8 (IL-8), interleukin-10 (IL-10) and tumor necrosis factor-α (TNF-α) have to be considered [10]. IL-6, IL-1, TNF-α and monocyte chemoattracting protein (MCP-1) are important mediators, capable of inducing inflammatory renal tubular damage through recruitment of macrophages and monocytes, as well as via activation of platelet aggregating factor (PAF) [11]. Renal ischemia may in turn affect angiotensin converting enzyme (ACE) expression, and can promote pulmonary vascular permeability, thereby impairing the function of ion channels responsible for reabsorption of fluids at level of pulmonary alveoli [12]. The diagnosis of renal damage is essentially based on the evaluation of renal albumin excretion [13]. Other indicators of kidney damage are urinary sediment changes, blood test abnormalities, and imaging alterations [14]. In large study populations, a relevant parameter used to evaluate the deterioration of renal function is the estimated glomerular filtration rate (eGFR), whose variability correlates with a poor prognosis [15].

Indeed, among COPD patients the concomitant occurrence of CKD exerts a negative impact on both global health status and overall mortality [16,17,18]. In particular, recent studies suggest that in COPD patients with CKD and hyperuricemia, uric acid (UA) levels may be an independent predictor of exacerbations and early mortality [19,20,21]. However, the pathophysiological relationship between COPD and UA is not yet fully understood [22]. Of note, UA levels above 6.9 mg/dL are an independent predictor of mortality at 30 days since the onset of a COPD exacerbation, but not at one year [19]. In addition, patients with high UA levels require longer hospitalizations, noninvasive ventilation, and more frequent admissions to the intensive care unit in the next 30 days [19]. Moreover, elevated plasma UA concentrations are associated with an increased risk of rehospitalization for COPD exacerbations at one year [19]. Interestingly, the Takahata study suggests that a significant inverse correlation exists between serum UA levels and forced vital capacity (FVC), as well as between UA and forced expiratory volume in the first second (FEV_1_) [23]. Consistent with this observation, hyperuricemia was found to be associated with a worse lung function and a higher risk of respiratory symptoms in COPD, even if a direct causal relationship has not been shown [24]. Finally, further evidence indicates that not only the measurement of serum UA levels, but also the relationship between UA and creatinine can be reliable to predict the risk of clinical worsening and exacerbation in COPD patients [25]. It is thus possible to speculate that high UA levels, by favoring CKD development, may negatively impact on clinical outcome in COPD patients.

In light of the above considerations, the main purpose of our study was to investigate the predictive factors of CKD development in COPD patients, with particular regard to the role of UA. The secondary aim focused on the search, in the same study population, for factors associated with the rapid decline of eGFR.

## 2. Methods

### 2.1. Study Population

In the present multicenter prospective observational study, 707 Caucasian subjects (367 males and 340 females) with COPD were recruited at the Geriatrics (558 patients) and Respiratory (68 patients) Units of “Magna Graecia” University Hospital of Catanzaro, Italy, and at the First Internal Medicine Unit (81 patients) of Sapienza University of Rome, Italy. The diagnosis of COPD was made according to the current Global Initiative for Chronic Obstructive Lung Disease (GOLD) recommendations [26]. Spirometry was performed according to the American Thoracic Society (ATS)/European Respiratory Society (ERS) guidelines [27]. All recruited COPD patients (GOLD stages 1 and 2) were monitored through a follow-up period of at least one year. Moreover, all enrolled patients had not yet started long-acting β_2_-adrenergic agonist (LABA), long-acting muscarinic receptor antagonist (LAMA) or inhaled corticosteroid (ICS) therapy because COPD had just been diagnosed. However, patients with active cancer, advanced hepatic disease, severe renal failure (baseline eGFR < 30 mL/min/1.73 m^2^), and acute kidney injury were excluded from the study. Furthermore, other exclusion criteria were the presence of chronic inflammatory or systemic autoimmune diseases, treatment with xanthine oxidase inhibitors, as well as use of hormone replacement therapy in female patients within the past 3 months.

All enrolled patients underwent anamnestic data collection and medical examination. A full anthropometric assessment was performed with measurement of body weight, height and body mass index (BMI). All subjects had different cardiovascular risk factors, but none of them had developed clinical events. Patients with a systolic blood pressure (SBP) ≥140 mmHg and/or diastolic blood pressure (DBP) ≥90 mmHg were considered to be hypertensive according to current guidelines [28]. We also calculated pulse pressure (PP) values, defined as the difference between SBP and DBP values. Type 2 diabetes mellitus was defined according to the American Diabetes Association (ADA) criteria: glycated hemoglobin ≥6.5% (48 mmol/mol), fasting plasma glucose (FPG) ≥126 mg/dL (7 mmol/L), 2 h postload glucose ≥200 mg/dL (11.1 mmol/L) or use of glucose-lowering medication [29]. At the time of enrolment, disease duration did not exceed five years and no patient was on insulin treatment.

All laboratory tests were carried out after a fasting period of at least 12 h. Glycemia was measured by the glucose oxidation method (Beckman Glucose Analyzer II; Beckman Instruments, Milan, Italy). The concentrations of triglycerides, total cholesterol, low-density lipoprotein (LDL) and high-density lipoprotein (HDL), were measured by enzymatic method (Roche Diagnostics GmbH, Mannheim, Germany). Plasma concentration of insulin was determined by an assay based on chemiluminescence (Roche Diagnostics). Insulin sensitivity was estimated using the homeostasis model assessment (HOMA) index. Serum levels of high-sensitivity C-reactive protein (hs-CRP) levels were measured by turbidimetric immunoassay (Behring, Marburg, Germany). Serum creatinine and UA levels were determined by an automated technique based on the Jaffé chromogen measurement and by the URICASE/POD method (Boehringer Mannheim, Mannheim, Germany), combined in an automated analyzer. eGFR was calculated using the equation proposed by the Chronic Kidney Disease Epidemiology Collaboration (CKD-EPI) [30].

Patients were stratified according to the classification of “Kidney Disease: Improving Global Outcomes” (KDIGO) clinical practice guidelines [31], based on the following parameters: normal eGFR (>90 mL/min/1.73 m^2^, stage G1), mild reduction of eGFR (60–89 mL/min/1.73 m^2^, stage G2), moderate reduction of eGFR (30–59 mL/min/1.73 m^2^, stage G3), and severe reduction of renal function eGFR (<30 mL/min/1.73 m^2^, stage G4). Within an outpatient context, a second measurement of serum creatinine was performed at follow-up in a time window ranging from 3 to 5 years with respect to baseline.

This multicenter study satisfied the standards of good clinical practice (GCP) and the principles of the Declaration of Helsinki. All enrolled participants signed a written informed consent. The present observational investigation was approved by the local Ethical Committee of Calabria Region (code protocol number 2012.63).

### 2.2. Statistical Analysis

Data were expressed as mean ± standard deviation (SD) for continuous variables, and as frequencies and percentages for nominal data. Because the distribution of the studied variables was normal, Student’s *t*-test for unpaired data was used to compare continuous variables between groups and χ^2^ test was used for nominal data. The annual deterioration of eGFR, expressed as the difference between eGFR measurements at follow-up and baseline, divided by the number of years of follow-up, was calculated. Study population was subdivided according to the presence of CKD at baseline, and on the basis of either presence or absence of a rapid decline of eGFR (>5 mL/min/1.73 m^2^/year), with the aim of evaluating the features of subjects with renal function decline. Moreover, we carried out a multivariate logistic regression analysis to define the adjusted odds ratio (OR) of variables affecting the annual rapid deterioration in renal function. In the multivariate analysis, all factors potentially influencing kidney function were included (age, BMI, smoking, gender, serum UA, diabetes, LDL-cholesterol, baseline eGFR, HOMA index, PP, hs-CRP, pharmacological treatment with RAAS inhibitor drugs, calcium channel blockers, diuretics, other antihypertensive drugs, statins, antiplatelet agents, inhaled treatments for COPD and exacerbations for the disease). In addition, the occurrence of CKD (eGFR < 60 mL/min/1.73 m^2^) at follow-up was measured as number of cases per 100 patient-year. Moreover, considering that the second serum creatinine value was not measured at the same time for all subjects in the follow-up, a regression analysis based on the Cox proportional hazard model was used to calculate the relative corrections of the factors associated with eGFR <60 mL/min/1.73 m^2^. We consecutively performed univariate and multivariate Cox regression analyses to calculate the hazard ratio (HR) of risk factors associated with the incidence of CKD. The multivariate analysis was carried out including all the variables that were significant according to univariate testing.

A p value lower than 0.05 was considered to be statistically significant. All analyses were performed using the SPSS 20.0 statistical program for Windows (SPSS Inc., Chicago, IL, USA).

## 3. Results

Among the 707 patients (48.1% women) enrolled in this study, the mean age was 61.8 ± 9.9 years, the mean e-GFR values were 79.7 ± 26.1 mL/min/1.73 m^2^ and 157 (22.2%) patients had renal dysfunction at baseline (eGFR <60 mL/min/1.73 m^2^). The anthropometric, hemodynamic, and biohumoral characteristics of the entire study population according to the presence or absence of renal dysfunction at baseline are reported in Table 1. No statistically significant differences were evident between the two groups with regard to gender, BMI, SBP, DBP, triglycerides, total and LDL-cholesterol. Participants with renal dysfunction at baseline were older and showed significantly higher values of PP, fasting glucose, insulin, HOMA, UA and hs-CRP, and lower values of HDL-cholesterol and pre-bronchodilator FEV_1_ (Table 1). The mean post-bronchodilator FEV_1_ value was 83.4 ± 23.5% predicted (pred.). Moreover, the mean baseline FVC and FEV_1_/FVC values were 91.4 ± 26.7% pred. and 0.61 ± 0.07, respectively. Table 2 shows the prevalence of the different cardiovascular risk factors and drug treatments in the entire study population according to the presence or absence of renal dysfunction. In the whole study population, 61% of patients were smokers and 75.7% hypertensive, 35.5% were obese, 36.6% had hypercholesterolemia, and 26% suffered from diabetes. At baseline, 82% were taking antihypertensive drugs, of which 66.3% were receiving renin angiotensin aldosterone system (RAAS) inhibitors, 34.6% calcium antagonists, and 39.6% diuretics. Moreover, 22.9% of patients utilized antiplatelet agents, 23.9% were on treatment with statins, and 49.5% were taking oral antidiabetic drugs. Among CKD patients there was a significantly higher number of diabetic subjects and a higher percentage of patients on oral antidiabetic therapy (Table 2). After COPD diagnosis at baseline, patients were treated with LABA/LAMA according to current guidelines, moreover, seventy exacerbations were observed during the follow-up (2.27%/100 patient-year).

### 3.1. Incident CKD (eGFR < 60 mL/min/1.73 m^2^)

Every patient underwent a second measurement of serum creatinine after a mean follow-up period of 52.3 ± 30.2 months. Figure 1 shows the distribution of the different eGFR classes in the entire cohort of COPD patients at baseline and during follow-up. Considering a cut-off value of 60 mL/min/m^2^, during follow-up one hundred patients developed CKD (2.38%/100 patient-year). In particular, the percentage of patients with eGFR >90 mL/min/1.73 m^2^ decreased from 29.3% (baseline) to 19.4% (follow-up). The percentage of patients with eGFR between 60 and 89 mL/min/1.73 m^2^ changed from 48.5 to 44.3%. Moreover, the proportion of patients with eGFR between 30 and 59 mL/min/1.73 m^2^ increased from 22.2 to 34.9%. At follow-up, 1.41% of patients developed severe renal failure <30 mL/min/1.73 m^2^ (Figure 1).

By performing univariate Cox regression analysis, a statistically significant association was found between CKD incidence and female gender, serum UA, diabetes, and baseline eGFR (Table 3—Panel A). In contrast, there were no statistically significant associations between CKD incidence and age, BMI, smoking, HOMA index, PP, hs-CRP, therapy with RAAS inhibitor drugs, calcium channel blockers, diuretics, other antihypertensive drugs, statins, antiplatelet agents, baseline FEV_1_, inhaled treatments for COPD and exacerbations for the disease (Table 3—Panel A). Factors associated with CKD in the univariate Cox analysis were included into a multivariate Cox analysis model, in order to determine the independent predictors of CKD (Table 3—Panel B). By this analysis, it was shown that UA (HR 1.148, *p* < 0.0001) and diabetes (HR 1.050, *p* < 0.0001) were the strongest predictors of CKD occurrence in COPD patients (Table 3—Panel B).

### 3.2. Rapid Decline of Renal Function (>5 mL/min/1.73 m^2^/Year)

Table 4 shows the features of COPD patients according to rapid deterioration in renal function (>5 mL/min/1.73 m^2^/year). Overall, 200 (28.2%) patients showed a reduction of eGFR >5 mL/min/1.73 m^2^/year. No statistically significant differences were found for BMI, DBP, total cholesterol, LDL cholesterol, HDL cholesterol, triglycerides, and hs-CRP. In contrast, there was a significantly higher prevalence of females among the patients with a rapid decline in eGFR, they were older and presented significantly higher values of SBP, PP, fasting glucose, insulin and HOMA, uric acid, baseline eGFR and lower values of FEV_1_ (Table 4).

The prevalence of different cardiovascular risk factors and current therapies, according to the rapid decline of renal function (>5 mL/min/1.73 m^2^/year), is shown in Table 5. In the group of patients characterized by a rapid decline of renal function, we observed a higher prevalence of diabetes and a greater percentage of subjects under antihypertensive treatment. No statistically significant differences were found in the prevalence of subjects who were smokers, hypertensive, obese, hypercholesterolemic, and treated with antiplatelet or antidiabetic drugs (Table 5).

A univariable logistic regression analysis of rapid deterioration in renal function (>5 mL/min/1.73 m^2^/year) was performed (Table 6—Panel A). Pulse pressure, UA, diabetes and baseline eGFR were significantly associated with the appearance of rapid renal deterioration. In particular, an increase of 1 mg/dl of UA was associated with a more than double risk of rapid decline in renal function (OR 2.296, *p* = 0.001). In addition, the presence of diabetes was associated with a 10.2% increased risk of rapid renal function decline while a 10 mL/min/1.73 m^2^ increase in baseline eGFR was associated with a 5.5% risk. Finally, a 10 mmHg of PP increase justified a 2.3 higher risk of rapid renal function deterioration. No statistically significant associations were found with regard to age, BMI, smoking, HOMA, hs-CRP, therapy with RAAS-inhibiting drugs, calcium channel blockers, diuretics, other antihypertensive drugs, statins, antiplatelet agents, and baseline FEV_1_ (Table 6—Panel A). The variables significantly associated with a rapid deterioration of renal function, according to the univariable logistic analysis, were included in a multivariable logistic regression analysis model, to reveal independent predictors of rapidly declining renal function (Table 6—Panel B).

This last analysis documented that, even if the association was attenuated, UA was the strongest predictor of rapid decline in renal function. In particular, an increase of 1 mg/dL of UA was associated with a double risk of rapid deterioration of renal function (OR 2.158, *p* < 0.0001). Diabetes and baseline eGFR were also significantly retained in the model (Table 6—Panel B).

## 4. Discussion

The present study shows that UA is associated with a rapid decline in kidney function and with an enhanced risk of incident kidney disease in COPD patients included within GOLD stages 1 and 2. A baseline renal dysfunction was quite common in the COPD patients enrolled in our study. Indeed, 77.8% exhibited normal renal function, whereas 22.2% of the recruited participants had an eGFR < 60 mL/min/1.73 m^2^. Similar to our findings, 20% and 31% of COPD patients recruited in previous studies complained of overt CKD [32,33], whereas a further investigation reported a CKD prevalence in COPD patients of only 9.6% in females and 5.1% in males [34]. However, these apparent discrepancies might be explained by the impact of both genetic and environmental factors on different ethnic populations living in distant geographic areas. During follow up, 100 out of 707 subjects developed CKD and 200 out of 707 presented a rapid decline of renal function. When we investigated the factors associated with renal dysfunction, we found that serum UA levels and diabetes were associated with both incident CKD and rapid decline of the eGFR. In particular, a 1 mg/dL increase in serum UA levels enhanced the risk of CKD incidence by 14.8% and 1 mg/dL increment in serum UA levels significantly enhanced the risk of experiencing a rapid deterioration of renal function during follow-up. In addition, the presence of diabetes increased the risk of developing renal dysfunction by 5%. Indeed, older patients with type 2 diabetes are susceptible to developing renal impairment and albuminuria [35]. However, these two pathological conditions are not necessarily concomitant in diabetic subjects, thus suggesting that different risk factors may possibly lead to either kidney failure or albuminuria [36]. Anyway, it is very important to monitor renal function in diabetic subjects, in order to detect as early as possible such comorbidities and to slow down the progressive deterioration of kidney performance. It is also noteworthy that COPD patients are more susceptible than the general population to manifest CKD [5,6]. The development of CKD in COPD patients negatively affects health status and causes mortality [16,17,18]. Interestingly, in COPD patients who complain of both CKD and hyperuricemia, UA levels can be considered a quite reliable predictive factor of exacerbation and early mortality [19,20,21].

In COPD patients, the association between high serum UA levels and CKD incidence can be explained by several pathophysiological mechanisms, that correlate UA with inflammation and endothelial dysfunction [37]. For instance, by reacting with peroxynitrite, UA has been shown to improve many pathologic situations that are associated with an amplified oxidative stress [38]. On the other hand, however, the reaction between UA and peroxynitrite itself generates free radicals and promotes the potential pro-oxidant activity of UA [38]. This latter event can occur when high values of UA are associated with diseases that underlie conditions of increased oxidative stress, such as obesity, metabolic syndrome, kidney disease, cardiovascular disease, and also COPD [39]. Some studies performed in hyperuricemic rats have shown that high levels of UA were associated with reduced plasma nitric oxide (NO) concentrations and altered urinary excretion of end products of NO catabolism [40,41]. By affecting vascular endothelial function, hyperuricemia can cause vasoconstriction of kidney vessels, arterial and glomerular hypertension, and preglomerular arteriolopathy [42]. In particular, increased oxidative stress is involved in the mechanisms by which UA limits intrarenal NO availability [43]. Furthermore, using an animal model of hyperuricemia, Mazzali et al. demonstrated that increased UA levels were associated with systemic arterial hypertension and renal ischemic damage, characterized by collagen deposition, macrophage infiltration, and tubular increase of osteopontin [44]. A further molecular mechanism by which UA is able to induce renal damage and arterial hypertension seems to be related to UA capability of activating RAAS system and decreasing NO levels [45]. Consistently with such findings, in hyperuricemic rats a decreased expression of nitric oxide synthase 1 (NOS1) in the macula densa, as well as an increased renin concentration in the juxtaglomerular system were detected [44]. Enalapril and L-arginine administration were able to prevent arterial hypertension and renal damage [44].

In COPD subjects, a significant increase in chronic oxidative stress levels was observed [10], together with a concomitant decrease in the production of antiaging molecules such as sirtuins [46]. In addition to COPD, molecular pathways leading to cellular senescence and inflammaging can indeed be activated also in chronic cardiovascular, metabolic and neurodegenerative disorders [47,48,49]. In these subjects oxidative stress is induced not only by cigarette smoke, but also by the activation of inflammatory cells such as neutrophils and macrophages [8]. As a consequence, the oxidative stress environment occurring in patients with COPD offers a fertile ground for a shift towards the pro-oxidant effect of UA, which contributes to both endothelial damage and progressive renal dysfunction [50]. Moreover, cigarette smoking exerts an indirect pathogenic action by injuring pulmonary vascular endothelium. A similar endothelial damage can be also induced in mouse models at the level of renal vessels [51]. Oxidative stress also leads to an enlarged synthesis of advanced glycation end products (AGEs), which in turn activate their receptors (RAGEs) [52]. This pathway seems to be upregulated in both COPD and diabetic nephropathy [52]. Consistently, our results also show that diabetes was associated with the onset of renal dysfunction. Finally, our study has made it possible to observe that COPD patients display a quite severe insulin resistance, resulting from ongoing systemic inflammation. A direct consequence of insulin resistance includes the reduction of circulating levels of IGF-1, a mediator that exerts positive effects on renal hemodynamics, especially with regard to its capability of incrementing renal blood flow and glomerular filtration rate by stimulating the L-arginine oxidase pathway [53]. Therefore, decreased IGF-1 levels have a negative impact on eGFR. However, we did not measure serum IGF-1 levels.

On the basis of such considerations, the enhanced mortality rate observed by Zhang et al. in hyperuricemic patients with COPD, could be attributable to a reduction of eGFR, which is known to increase the global cardiovascular risk [21]. In COPD patients, the association between high UA and creatinine levels underlies a tendency to experience a faster worsening of respiratory symptoms and lung function [24]. This implies that in COPD different pathogenic pathways, leading to pulmonary, renal and metabolic dysfunctions converge to produce an overall deterioration in patient outcomes. In addition, Rumora et al. suggested that UA might be a useful biomarker when combined with IL-1β, whereas the UA/creatinine ratio might be even more informative with regard to the global evaluation of COPD patients [54].

## 5. Limitations

The limitations of this study include its observational design and the lack of data during follow-up with regard to glycemic control, as well as to some relevant COPD severity assessments such as CAT score, lung function decline and DLCO. Another relevant limitation of our study regards the lack of proteinuria levels. Furthermore, the renal function impairment was assessed by using only a second measurement of serum creatinine during the follow-up period, whereas the monitoring of creatinine serum concentrations more than twice would be better for the detection of renal function deterioration. Moreover, we included only adult Caucasian patients, so that the generalizability of our results to other cardiovascular clinical settings or to other ethnicity groups is uncertain. Finally, we excluded patients with severe COPD.

## 6. Conclusions

In conclusion, the interesting findings of our present study, referring to a large number of COPD patients, suggests that UA represents an independent predictor of CKD occurrence and rapidly progressive decline in renal function. Overall, the disease association including COPD, hyperuricemia and kidney failure persists even after correction for the known risk factors involved in renal damage (Figure 2). Therefore, the evaluation of serum UA levels and treatment of hyperuricemia play crucial roles in the management of COPD patients belonging to functional GOLD stages 1 and 2. In particular, UA measurement could be suitable for the detection of COPD subjects who are at high risk of developing renal function deterioration, thus highlighting the importance of creatinine monitoring. Furthermore, the identification of COPD participants carrying a high risk of renal disease progression could be useful to detect those subjects who are more prone to developing cardiovascular events.

## Figures and Tables

**Figure 1 nutrients-13-02811-f001:**
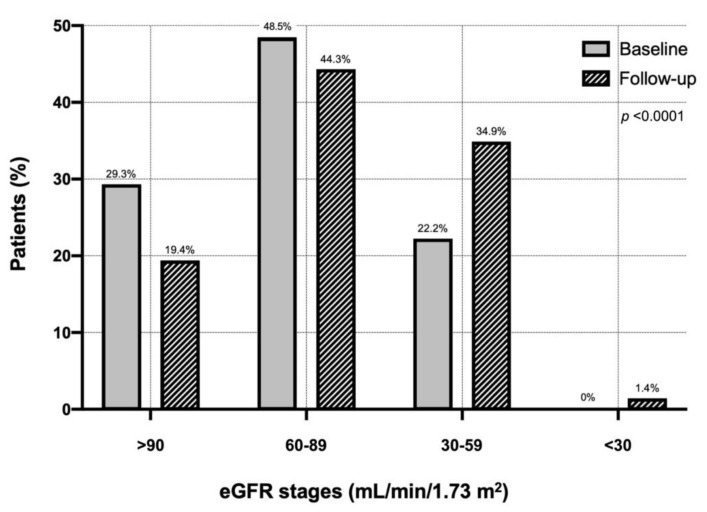
Distribution of eGFR classes in the entire cohort of chronic obstructive pulmonary disease patients at baseline and during follow-up. Abbreviations: eGFR, estimated glomerular filtration rate.

**Figure 2 nutrients-13-02811-f002:**
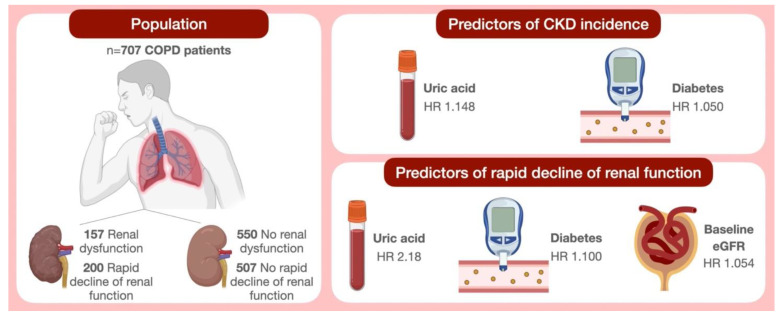
Schematic illustration of study findings. Abbreviations: COPD, chronic obstructive pulmonary disease; CKD, chronic kidney disease; eGFR, estimated glomerular filtration rate; HR, hazard ratio.

**Table 1 nutrients-13-02811-t001:** Anthropometric, hemodynamic, and biohumoral characteristics, stratified according to the presence or absence of renal dysfunction (eGFR < 60 mL/min/1.73 m^2^) at baseline.

	All	Renal Dysfunction	No Renal Dysfunction	*p*
	(*n* = 707)	(*n* = 157)	(*n* = 550)	
eGFR, mL/min/1.73 m^2^	79.7 ± 26.1	48.3 ± 8.5	88.6 ± 22.2	<0.0001
Gender, m/f	367/340	88/69	279/271	0.239
Age, years	61.8 ± 9.9	67.4 ± 5.7	60.2 ± 10.3	<0.0001
BMI, kg/m^2^	29.2 ± 5.8	29.1 ± 5.9	29.2 ± 5.7	0.859
SBP, mmHg	141.2 ± 15.4	143.5 ± 18.6	140.5 ± 14.3	0.065
DBP, mmHg	82.7 ± 10.5	82.8 ± 11.3	82.7 ± 10.3	0.970
Pulse pressure, mmHg	58.4 ± 14.3	60.7 ± 16.1	57.8 ± 13.9	0.038
Fasting glucose, mg/dL	103.7 ± 22.8	115.5 ± 34.4	100.4 ± 16.9	<0.0001
Fasting insulin, mU/mL	12.6 ± 5.8	15.8 ± 8.5	11.7 ± 4.5	<0.0001
HOMA	3.2 ± 1.9	4.4 ± 3.1	2.9 ± 1.2	<0.0001
Total cholesterol, mg/dL	198.8 ± 46.1	200.9 ± 40.3	198.1 ± 47.6	0.489
LDL-cholesterol, mg/dL	117.1 ± 35.6	117.4 ± 36.3	117.1 ± 35.4	0.919
HDL-cholesterol, mg/dL	48.9 ± 14.8	45.2 ± 11.8	50.1 ± 15.4	<0.0001
Triglycerides, mg/dL	137.3 ± 87.9	141.2 ± 97.8	136.1 ± 84.9	0.554
Creatinine, mg/dL	1.01 ± 0.2	1.21 ± 0.4	0.96 ± 0.1	<0.0001
Uric acid, mg/dL	5.5 ± 2.5	6.9 ± 2.4	5.1 ± 2.3	<0.0001
hs-CRP, mg/dL	2.9 ± 2.1	3.6 ± 2.4	2.7 ± 1.9	<0.0001
Pre-bronchodilator FEV_1_, %	82.3 ± 22.2	81.7 ± 22.3	86.2 ± 21.6	0.022
FEV_1_/FVC	0.61 ± 0.07	0.60 ± 0.07	0.62 ± 0.06	0.013

Abbreviations: eGFR, estimated glomerular filtration rate; BMI, body mass index; SBP, systolic blood pressure; DBP, diastolic blood pressure; HOMA, homeostatic model assessment; LDL, low-density lipoprotein; HDL, high-density lipoprotein; hs-CRP, high-sensitivity C-reactive protein; FEV_1_, forced expiratory volume in the 1st second; FVC, forced vital capacity.

**Table 2 nutrients-13-02811-t002:** Cardiovascular risk factors and therapies, stratified according to the presence or absence of renal dysfunction (eGFR < 60 mL/min/1.73 m^2^) at baseline.

	All	Renal Dysfunction	No Renal Dysfunction	*p*
	(*n* = 707)	(*n* = 157)	(*n* = 550)	
Smokers, *n* (%)	436 (61)	101 (64.3)	330 (60)	0.326
Hypertension, *n* (%)	535 (75.7)	120 (76.4)	415 (75.4)	0.801
Obesity, *n* (%)	251 (35.5)	53 (33.8)	198 (36)	0.604
Hypercholesterolemia, *n* (%)	259 (36.6)	61 (38.8)	198 (36)	0.512
Diabetes, *n* (%)	184 (26)	52 (33.1)	132 (24)	0.021
Drugs				
Antihypertensives, *n* (%)	580 (82)	130 (82.8)	450 (81.8)	0.776
—RAAS inhibitors, *n* (%)	469 (66.3)	104 (80)	365 (81.1)	0.776
—Calcium-blockers, *n* (%)	245 (34.6)	55 (42.3)	190 (42.2)	0.986
—Diuretics, *n* (%)	280 (39.6)	70 (53.8)	210 (46.6)	0.149
—Others, *n* (%)	43 (6.1)	12 (9.2)	31 (6.9)	0.751
Antiplatelet agents, *n* (%)	162 (22.9)	40 (25)	122 (22.1)	0.386
Statins, *n* (%)	169 (23.9)	44 (28)	125 (22.7)	0.169
Oral antidiabetic drugs, *n* (%)	350 (49.5)	90 (57.3)	260 (47.3)	0.026

Abbreviations: eGFR, estimated glomerular filtration rate; RAAS, renin angiotensin aldosterone system.

**Table 3 nutrients-13-02811-t003:** Univariable (Panel A) and multivariable (Panel B) Cox regression analysis of chronic kidney disease incidence (eGFR < 60 mL/min/1.73 m^2^).

**Panel A**	**Hazard Ratio**	**95% Confidence Interval**	***p***
Sex, m/f	0.649	0.492–0.855	0.002
Uric acid, 1 mg/dL	1.153	1.099–1.210	<0.0001
Diabetes, yes/no	1.09	1.022–1.304	<0.0001
Baseline eGFR, 10 mL/min/1.73 m2	0.968	0.960–0.975	<0.0001
**Panel B**	**Hazard Ratio**	**95% Confidence Interval**	***p***
Uric acid, 1 mg/dL	1.148	1.098–1.201	<0.0001
Diabetes, yes/no	1.050	1.019–1.290	<0.0001

Abbreviations: eGFR, estimated glomerular filtration rate.

**Table 4 nutrients-13-02811-t004:** Anthropometric, hemodynamic, and biohumoral characteristics, stratified according to the presence or absence of rapid decline of renal function (>5 mL/min/1.73 m^2^/year).

	Rapid Decline of eGFR	No rapid Decline of eGFR	*p*
	(*n* = 200)	(*n* = 507)	
Sex, m/f	87/113	280/227	0.004
Age, years	62.3 ± 9.4	60.4 ± 11.1	0.034
BMI, kg/m^2^	29.4 ± 6.1	29.1 ± 5.6	0.603
SBP, mmHg	142.3 ± 15.4	138.6 ± 14.9	0.005
DBP, mmHg	82.6 ± 10.7	83.1 ± 10.1	0.637
Pulse pressure, mmHg	59.5 ± 14.6	55.6 ± 13.2	0.001
Fasting glucose, mg/dL	105.5 ± 24.9	99.3 ± 15.4	<0.0001
Fasting Insulin, mU/mL	12.9 ± 6.1	11.5 ± 5.1	0.001
HOMA	3.4 ± 2.1	2.8 ± 1.4	<0.0001
Cholesterol, mg/dL	199.9 ± 37.2	198.4 ± 48.4	0.694
LDL, mg/dL	121.7 ± 40.1	115.4 ± 33.6	0.063
HDL, mg/dL	48.4 ± 14.6	50.5 ± 15.2	0.103
Hypercholesterolemic, mg/dL	137.9 ± 87.1	135.6 ± 90.1	0.801
Creatinine, mg/dL	0.97 ± 0.1	1.01 ± 0.3	<0.0001
eGFR, mL/min/1.73 m^2^	93.8 ± 24.8	74.1 ± 24.4	<0.0001
Uric acid, mg/dL	7.5 ± 2.2	4.7 ± 2.1	<0.0001
hs-CRP, mg/dL	2.7 ± 2.1	2.9 ± 2.2	0.180
Pre-bronchodilator FEV_1_, %	79.8 ± 21.2	83.8 ± 22.5	0.032
FEV_1_/FVC	0.60 ± 0.08	0.62 ± 0.07	0.019

Abbreviations: eGFR, estimated glomerular filtration rate; BMI, body mass index; SBP, systolic blood pressure; DBP, diastolic blood pressure; HOMA, homeostatic model assessment; LDL, low-density lipoprotein; HDL, high-density lipoprotein; hs-CRP, high-sensitivity C-reactive protein; FEV_1_, forced expiratory volume in the 1st second; FVC, forced vital capacity.

**Table 5 nutrients-13-02811-t005:** Cardiovascular risk factors and therapies, stratified according to the presence or absence of rapid decline of renal function (>5 mL/min/1.73 m^2^/year).

	Rapid Decline of eGFR	No Rapid Decline of eGFR	*p*
	(*n* = 200)	(*n* = 507)	
Smokers, *n* (%)	123 (61.5)	308 (60.7)	0.853
Hypertension, *n* (%)	145 (72.5)	390 (76.9)	0.217
Obesity, *n* (%)	72 (36)	179 (35.3)	0.862
Hypercholesterolemia, *n* (%)	77 (38.5)	182 (35.9)	0.517
Diabetes, *n* (%)	69 (34.5)	115 (22.7)	0.001
Drugs			
Antihypertensives, *n* (%)	175 (87.5)	405 (79.9)	0.017
—RAAS inhibitors, *n* (%)	139 (79.4)	330 (81.5)	0.564
—Calcium-blockers, *n* (%)	69 (39.4)	176 (43.4)	0.262
—Diuretics, *n* (%)	88 (50.3)	192 (47.4)	0.429
—Others, *n* (%)	18 (10.3)	25 (4.9)	0.203
Antiplatelet agents, *n* (%)	55 (27.5)	107 (21.1)	0.068
Statins, *n* (%)	57 (28.5)	113 (22.3)	0.081
Antidiabetics, *n* (%)	100 (50)	250 (49.3)	0.868

Abbreviations: eGFR, estimated glomerular filtration rate; RAAS, renin angiotensin aldosterone system.

**Table 6 nutrients-13-02811-t006:** Univariable (Panel A) and multivariable (Panel B) logistic regression analysis of the rapid decline of renal function (>5 mL/min/1.73 m^2^/year).

**Panel A**	**Odds Ratio**	**95% Confidence Interval**	***p***
Pulse pressure, 10 mmHg	1.023	1.011–1.060	0.001
Uric acid, 1 mg/dL	2.296	1.992–2.646	<0.0001
Diabetes, yes/no	1.102	1.097–1.324	0.011
Baseline eGFR, 10 mL/min/1.73 m^2^	1.055	1.043–1.068	<0.0001
**Panel B**	**Odds Ratio**	**95% Confidence Interval**	***p***
Uric acid, 1 mg/dL	2.158	1.896–2.457	<0.0001
Diabetes, yes/no	1.100	1.096–1.320	0.009
Baseline eGFR, 10 mL/min/1.73 m^2^	1.054	1.043–1.066	<0.0001

Abbreviations: eGFR, estimated glomerular filtration rate.

## Data Availability

The data presented in this study are available on request from the corresponding author.

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
