# Peer review of "Predictors of Renal Function Worsening in Patients with Chronic Obstructive Pulmonary Disease (COPD): A Multicenter Observational Study"

_nutrients, 2021, doi:10.3390/nu13082811_

Round 1
Reviewer 1 Report
The manuscript titled as "PREDICTORS OF RENAL FUNCTION WORSENING IN PATIENTS WITH CHRONIC OBSTRUCTIVE PULMONARY DISEASE (COPD): A MULTICENTER OBSERVATIONAL STUDY. " This manuscript is a multicentre prospective observational study of 707 patients with COPD.
Comments and Suggestions for Authors
The paper is overall well structured despite a some comments. I have suggestion that could improve the paper:
- Introduction
- The introduction explains the most important information about the study.
- Material and methods:
- Student’s t-test was used to the statistical analysis, please explain whether the distribution of the studied variables was normal?
- Results:
- Figure 1 shows the distribution of the different eGFR classes in the entire cohort of COPD patients at baseline and during follow-up. please add p-value between baseline and during follow-up.
- Discussion
- In my opinion the discussion is well written.
Author Response
We would like to thank very much the Reviewers for having carefully reviewed our manuscript, thus suggesting those changes which have significantly improved its overall quality. We have prepared a revised version of this paper, taking into account the comments of the reviewers. We also include a point-by-point response to the criticisms raised by the Referees. The changes have been highlighted in yellow, and deletions have been evidenced in red. Therefore, we hope that our manuscript is now suitable for publication in Nutrients.
REVIEWER 1
The paper is overall well structured despite some comments. I have suggestion that could improve the paper:
Introduction:
The introduction explains the most important information about the study.
Material and methods:
Student’s t-test was used to the statistical analysis, please explain whether the distribution of the studied variables was normal?
The data distribution was normal, so we used the Student’s t-test. This concept has been specified in the revised text (lines 145-146).
Results:
Figure 1 shows the distribution of the different eGFR classes in the entire cohort of COPD patients at baseline and during follow-up. please add p-value between baseline and during follow-up.
P-value has been added in the revised figure 1 (line 217).
Discussion
In my opinion the discussion is well written.

Reviewer 2 Report
Thank you for giving me the opportunity to review this manuscript.
This is a well-written paper.
This paper has some interesting findings along with a good structure.
However, some limitations in terms of study design and methodology should be addressed before the decision of acceptance.
I add the following points to revise.
Major revisions
1. The authors wrote in abstract and discussion of this paper that COPD patients have a significant worsening of renal function over time ( abstract, line no 31), and overall the disease association including COPD, hyperuricemia and kidney failure persists even after correction for the known risk factors involved in renal damage (discussion line no 379-381).
In table 3 and table 6, FEV1 is not associated with the rapid decline of renal function, although only table 4 shows rapid decliners of GFR have lower lung function.
Therefore, I cannot understand how authors can say that COPD patients have a significant worsening of renal function over time by the presented data.
2. It is well known that elevated serum uric acid level and diabetes are predictors for the development of chronic kidney disease, so there is nothing new in those findings.
Because this paper was conducted with COPD cohort, authors should analyze what COPD related factors including acute exacerbation of COPD, CAT score, lung function decline, DLCO, FEV1/FVC, FVC, etc influenced or related to the primary outcome (renal function decline) of this study.
3. The authors wrote that considering that the second serum creatinine value was not measured at the same time for all patients in the follow-up, a regression analysis based on the Cox proportional hazard model was used to calculate the relative corrections of the factors associated with eGFR (line 162 -164).
But this description doesn’t seem to be enough for validation.
Because second serum creatinine is a primary outcome of this study, the method of obtaining it should be more specified.
Was follow-up serum creatinine measured based on the protocol or any given timeframe with some window period?
If the follow-up serum creatinine had been measured at hospitalization accompanied by conditions causing acute renal failure, the misleading data could have been collected.
4. Table 1 shows that the renal dysfunction group had older age.
Were the multivariable analyses adjusted for age?
5. The method says that COPD diagnosis was made according to GOLD.
However, Table 1 and 4 show only FEV1, which is not clear whether it is pre bronchodilator or post bronchodilator value. Basic pulmonary function data including post bronchodilator FEV1 data along with FEV1/ FVC and FVC should be provided in this paper.
Minor revision
1. This is a multicenter observational study. Method segment should include how many patients were enrolled at each participating institution.
Author Response
We would like to thank very much the Reviewers for having carefully reviewed our manuscript, thus suggesting those changes which have significantly improved its overall quality. We have prepared a revised version of this paper, taking into account the comments of the reviewers. We also include a point-by-point response to the criticisms raised by the Referees. The changes have been highlighted in yellow, and deletions have been evidenced in red. Therefore, we hope that our manuscript is now suitable for publication in Nutrients.
REVIEWER 2
Thank you for giving me the opportunity to review this manuscript.
This is a well-written paper.
This paper has some interesting findings along with a good structure.
However, some limitations in terms of study design and methodology should be addressed before the decision of acceptance.
I add the following points to revise.
Major revisions
- The authors wrote in abstract and discussion of this paper that COPD patients have a significant worsening of renal function over time (abstract, line no 31), and overall the disease association including COPD, hyperuricemia and kidney failure persists even after correction for the known risk factors involved in renal damage (discussion line no 379-381).
In table 3 and table 6, FEV1 is not associated with the rapid decline of renal function, although only table 4 shows rapid decliners of GFR have lower lung function. Therefore, I cannot understand how authors can say that COPD patients have a significant worsening of renal function over time by the presented data.
Many thanks for this interesting observation. We premise that all enrolled patients were affected by COPD (GOLD stages 1 and 2). As reported in the paragraph "Statistical analysis" (lines 155-159), in the multivariate logistic analysis of the rapid decline of renal function, all variables potentially affecting renal function were inserted (age, BMI, smoking, gender, serum UA, diabetes, LDL-cholesterol, baseline eGFR, HOMA index, PP, hs-CRP, pharmacological treatment with RAAS inhibitor drugs, calcium channel blockers, diuretics, other antihypertensive drugs, statins, antiaggregants, inhaled treatments for COPD and exacerbations for the disease). In particular, COPD therapy and the number of exacerbations were also included. We did not include FEV1 to avoid any collinearity in the model.
The entire cohort of COPD study patients had a deterioration of renal function over time according to figure 1 (lines 208-216). Furthermore, table 4 shows the clinical characteristics of COPD patients according to rapid decline in renal function (> 5 mL / min / 1.73 m2 / year); 28.2% of the overall population showed a reduction of eGFR> 5 mL / min / 1.73 m2 / year.
- It is well known that elevated serum uric acid level and diabetes are predictors for the development of chronic kidney disease, so there is nothing new in those findings. Because this paper was conducted with COPD cohort, authors should analyze what COPD related factors including acute exacerbation of COPD, CAT score, lung function decline, DLCO, FEV1/FVC, FVC, etc influenced or related to the primary outcome (renal function decline) of this study.
Many thanks for this comment. The main novelty of our study refers to the demonstration of the relevant role of uric acid and diabetes as predictors of renal function deterioration, within the context of the study population consisting of COPD patients (GOLD stages 1 and 2). As reported in the paragraph "Statistical analysis" (line 158-159), in the multivariate model we included as potential predictors of renal function worsening both COPD exacerbation number and inhaled therapy, which however did not result to be statistically significant. As suggested by the reviewer, in the multivariate model we tried to include FEV1, which was not statistically significant. In regard to the other parameters that the reviewer suggests to be considered, unfortunately these were not available in the shared database. However, we included this concept as one of the study limitations (lines 371-372). Anyway, this observation turns out to be an interesting suggestion for the design of an eventual future study, to be carried out in a new cohort of COPD patients.
- The authors wrote that considering that the second serum creatinine value was not measured at the same time for all patients in the follow-up, a regression analysis based on the Cox proportional hazard model was used to calculate the relative corrections of the factors associated with eGFR (line 162 -164).
But this description doesn’t seem to be enough for validation. Because second serum creatinine is a primary outcome of this study, the method of obtaining it should be more specified. Was follow-up serum creatinine measured based on the protocol or any given timeframe with some window period? If the follow-up serum creatinine had been measured at hospitalization accompanied by conditions causing acute renal failure, the misleading data could have been collected.
During the follow-up, serum creatinine was measured in a time window ranging from 3 to 5 years with respect to baseline. This concept has been specified in the revised text (lines 137-138). All patients who had been hospitalized were not included in our study, and each creatinine value was measured within an outpatient context. These concepts have been specified in the revised text (lines 103, 136-137).
- Table 1 shows that the renal dysfunction group had older age.
Were the multivariable analyses adjusted for age?
Yes, the multivariable analyses were adjusted for age.
- The method says that COPD diagnosis was made according to GOLD.
However, Table 1 and 4 show only FEV1, which is not clear whether it is pre bronchodilator or post bronchodilator value. Basic pulmonary function data including post bronchodilator FEV1 data along with FEV1/ FVC and FVC should be provided in this paper.
Many thanks for this observation. FEV1 values showed in Tables 1 and 4 referred to pre-bronchodilator FEV1. This concept has been specified in the revised text (lines 181, 199, 246). Baseline pulmonary function data including post-bronchodilator FEV1, FVC, and FEV1/FVC values have been provided in the revised text (lines 180-183).
Minor revision
- This is a multicenter observational study. Method segment should include how many patients were enrolled at each participating institution.
The number of patients enrolled at each participating institution has been specified in the revised text (lines 92-94).

Round 2
Reviewer 2 Report
Authors answered my questions well.
I have no further comment regarding revision.
Thank you for giving me the opportunity to appreciate this manuscript.
Author Response
We would like to thank very much this Reviewer for his/her useful comments.